# Interaction between Boron and Other Elements in Plants

**DOI:** 10.3390/genes14010130

**Published:** 2023-01-03

**Authors:** Ying Long, Jiashi Peng

**Affiliations:** 1Department of Personnel, Hunan University of Science and Technology, Xiangtan 411201, China; 2School of Life and Health Sciences, Hunan University of Science and Technology, Xiangtan 411201, China; 3Hunan Key Laboratory of Economic Crops Genetic Improvement and Integrated Utilization, Hunan University of Science and Technology, Xiangtan 411201, China; 4Key Laboratory of Ecological Remediation and Safe Utilization of Heavy Metal-Polluted Soils, Hunan University of Science and Technology, Xiangtan 411201, China

**Keywords:** boron, elements interaction, mineral nutrient, toxic metal

## Abstract

Boron (B) is an essential mineral nutrient for growth of plants, and B deficiency is now a worldwide problem that limits production of B deficiency-sensitive crops, such as rape and cotton. Agronomic practice has told that balanced B and other mineral nutrient fertilizer applications is helpful to promote crop yield. In recent years, much research has reported that applying B can also reduce the accumulation of toxic elements such as cadmium and aluminum in plants and alleviate their toxicity symptoms. Therefore, the relation between B and other elements has become an interesting issue for plant nutritionists. Here we summarize the research progress of the interaction between B and macronutrients such as nitrogen, phosphorus, calcium, potassium, magnesium, and sulfur, essential micronutrients such as iron, manganese, zinc, copper, and molybdenum, and beneficial elements such as sodium, selenium, and silicon. Moreover, the interaction between B and toxic elements such as cadmium and aluminum, which pose a serious threat to agriculture, is also discussed in this paper. Finally, the possible physiological mechanisms of the interaction between B and other elements in plants is reviewed. We propose that the cell wall is an important intermediary between interaction of B and other elements, and competitive inhibition of elements and related signal transduction pathways also play a role. Currently, research on the physiological role of B in plants mainly focuses on its involvement in the structure and function of cell walls, and our understanding of the details for interactions between B and other elements also tend to relate to the cell wall. However, we know little about the metabolic process of B inside cells, including its interactions with other elements. More research is needed to address the aforementioned research questions in future.

## 1. Introduction

Boron (B) was identified by Warington as an essential trace mineral element for the growth and development of higher plants through experiments on *Vicia faba* in 1923 [1]. Over the past century, extensive research has been conducted on how plants absorb and transport B and how B is metabolized. Whether B is an essential mineral element for vascular plants has recently become a controversial topic again [2], but it is generally recognized by the academic community that B is an essential mineral element in the life process of plants [3,4].

Among the essential minerals, B has the smallest atomic number and different characteristics from others in terms of its roles in the physiology of plants. For example, B is not part of the chemical compositions of enzymes, and it does not affect the activity of enzymes by binding to them, nor does it participate in redox reactions by changing its own valence, as iron (Fe), sulfur (S), and molybdenum (Mo) do. B is present in the soil solution as forms of boric acid (H_3_BO_3_) and borate (B(OH)_4_^−^), which is in an equilibrium that depends on pH. At common soil pH values (5.5–7.5), the most plentiful form is the soluble undissociated B(OH)_3_ [5].

B mainly exists in the cell wall of plants. Particularly, almost all B is bound to the cell wall in a low-B environment [6]. Matoh et al. isolated a B–polysaccharide complex from the roots of *Raphanus sativus* and revealed that this complex is ubiquitous in plants as a dimer (dRG-II–B) formed by two rhamnogalacturonan II (RG-II) monomers cross-linked with one B through the apiose residues on its A chain [7,8]. With high stability in the physiological pH range, dRG-II–B is crucial for maintaining the structure and function of the cell wall. B can conjugate with RG-II on the cell wall and glycosyl inositol phosphoryl ceramides (GIPCs) on the cell membrane to form a GIPC–B–RG-II complex [9]. Thus, B might play a role in cell–cell wall adhesion as well. Interestingly, B is also important for the growth of yeasts and bacteria, but its direct participation in the composition of their cell walls has not been uncovered, implying that B may be involved in other physiological processes of cells in addition to functioning in the cell wall.

B content varies widely between plants. In general, dicotyledonous plants, especially oilseed rape, cotton, and other crops, have a high demand for B and more easily suffer from B deficiency symptoms, such as decreased root elongation, dried-up floral buds, fewer pods and low seed yield or sterility in plant reproductive growth [10]. On the other hand, as an important abiotic stress in several agricultural areas worldwide, B toxicity also damages agricultural production along with the hindrance in root growth, necrosis on leaf tips and chlorosis [5,11,12]. Nonetheless, our limited knowledge about B in the metabolic process of plants is mainly about its role in the cell wall structure of plants [13], which has impeded the further development of B-efficient crop varieties. As a result, the problem of B deficiency in crop planting is still primarily solved by the application of B fertilizers. Thus, a good understanding of the interaction between B and other mineral nutrients would help guide the rationally combined fertilization of B with other mineral nutrients to increase nutrient utilization efficiency and avoid the excessive accumulation of toxic metal elements. Here we reviewed the research progress of the interaction between B and other 16 elements in plant, including nutrients, beneficial elements, as well as toxic metals. Furthermore, we summarized the possible mechanisms of the interaction between them and B.

## 2. Interaction of B with Other Mineral Nutrients

### 2.1. B and Nitrogen (N)

The rational combined application of N and B fertilizers has long been known to exert significant effects on the improvement of crop yields [14]. According to research on *Nicotiana tabacum*, under the condition of B deficiency, the nitrate concentrations in the leaves and roots of plants are low, which may be correlated with the downregulation of nitrate transporter gene *NRT2* and root plasma membrane H^+^-ATPase gene *PAM2* [15]. The concentration difference between the protons inside and outside the membrane, governed by the plasma membrane H^+^-ATPase, is essential for the transmembrane transport of nitrate ions. Therefore, the B deficiency-induced downregulation of *PAM2* expression might indirectly inhibit the transport of nitrate-N [15]. In *Zea mays* and *Brassica napus*, B is found to improve N utilization efficiency by boosting the synthesis of nitrate reductase [16,17]. Moreover, B has a critical influence on rhizobia-mediated N fixation. The role of B in plant symbiotic N fixation was first identified in the research on *Vicia faba* and has been verified in other legumes [18,19,20,21]. B is an essential player in maintaining the normal structure of the cell wall and membrane of the root nodule tissue. B can also affect interactions on the cell surface to promote the endocytosis of rhizobia by the host cells, thus participating in the host infection process of the bacteria [22].

Furthermore, the influences of B on N absorption and distribution vary with the N source. Ammonium salt and nitrate are two vital inorganic N sources. Plants excrete protons while absorbing ammonium salt, leading to reduced pH in the rhizosphere, and B mainly exists in the form of B(OH)_3_ in the rhizosphere soil. The pH in the rhizosphere is increased when plants absorb nitrate, and B in the rhizosphere soil is mainly converted to B(OH)_4_^−^ [23]. If B is abundant enough in the environment, B is mostly absorbed by plants via the free diffusion of boric acid. Hence, plants have lower B absorption when nitrate serves as the N source than when ammonium salt is the N source [24].

### 2.2. B and Phosphorus (P)

The B–P interaction is attractive to plant nutritionists, but existing research results still have no unified understanding of their relationship. According to research on tomato and maize, there is an antagonistic relationship between B absorption and P absorption. A possible reason is that phosphate and borate share the same absorption and transport system, which leads to a competitive relationship [25,26]. However, another study in maize has shown that B deficiency can aggravate various changes in physiological processes caused by both P deficiency and excess P toxicity, such as reduction in biomass and ribonuclease activity [27]. In addition, B absorption by *B. napus* has been found to be promoted by an appropriate amount of P, probably because (1) P can promote water absorption by plants and boost their growth and transpiration, and B absorption can be enhanced by the improvement of these physiological processes; and (2) P can influence the biochemical characteristics of plant rhizosphere, and hence raise the B availability in soil [28]. Under some extreme conditions of B deficiency or excess B toxicity, the impeded P absorption and transport to the aboveground part were observed in *V. faba* and Spanish groundnuts [29,30]. Similarly, the application of excess P fertilizers can reduce the B accumulation in mustard and tomato [25,31].

### 2.3. B and Calcium (Ca)

Through research on hydroponic *Solanum lycopersicum* in 1944, Reeve et al. found that a higher Ca concentration in the culture medium can aggravate the symptoms of B deficiency in plants, but alleviate the toxicity of high-concentration B stress [32]. On this basis, the concept of the Ca/B ratio was put forward. The Ca–B interaction has been verified by studies on numerous other plants [33,34,35]. Given the noticeable influence of the Ca–B interaction on B, the Ca/B ratio is commonly used to evaluate the abundance of B in plants, and the optimal Ca/B ratio for the growth of various plants is diverse. The Ca/B ratio is of key to the combined application of Ca and B fertilizers [36].

As for B, the cell wall is the largest Ca pool in plants as well. Ca^2+^ can combine with homogalacturonans (HG) in pectin to generate complexes [37]. Considering the vital roles of Ca and B in jointly stabilizing and maintaining the structure and function of the cell wall, the Ca–B interaction may have a close correlation with the physiological processes related to the cell wall. Moreover, B deficiency may alter the intracellular Ca^2+^ concentration and affect the expression of genes related to Ca^2+^ signal transduction. Hence, it is also argued that the influence of B on Ca^2+^ signal transduction could be an important manifestation of the Ca–B interaction [38]. Notably, the increased Ca concentration in soil usually reduces the B absorption by plants, possibly because the B availability in soil is lowered by the formation of calcium borate complexes [36].

### 2.4. B and Potassium (K), Magnesium (Mg), S

The interactions of B with K, Mg, and S in plants have not been researched much. Reeve et al. discovered, through a study on tomato, that the symptoms of B deficiency or excess B toxicity are further aggravated with rising K concentration [32]. In tobacco, K can promote the accumulation of B, which is more significant under a high concentration of B. Moreover, the elevated B concentration strengthens the accumulation of K in the roots and leaves in tobacco, but attenuates the accumulation of Mg in the tissues of tomato [39]. The opposite effects of B on the accumulation patterns of Mg and K may be attributed to the competitive relationship between Mg and K in the absorption mechanism [39,40]. However, no significant influence of B concentration on Mg accumulation is identified in maize [41]. An interaction also seems to exist between B and S. Specifically, S can significantly promote the B absorption by *Brassica juncea* and *Helianthus annuus*, and the S absorption is substantially facilitated by higher B concentrations [42]. Under the rationally combined application of B and S fertilizers, the biomass, seed yield, and protein and oil content of *B. juncea* and *H. annuus* increase [42], but the specific mechanism remains unknown.

### 2.5. B and Trace Elements

Fe accumulation in plants displays varying trends as the exogenously applied B concentration rises. When the B concentration is lower than the toxic concentration, Fe accumulation rises with the increased B concentration, while it exhibits an evident reduction in the case of excessive B concentration, suggesting that an appropriate amount of B may indirectly influence Fe absorption by promoting plant growth to boost the demand for Fe [39,41,43]. Recent research found that B also contributes to the long-distance transport of Fe from the roots to the aboveground part by promoting the reuse of apoplast Fe in the roots of *Arabidopsis thaliana* [44]. B influences manganese (Mn) accumulation similarly to Fe accumulation. In general, B promotes Mn accumulation in plants, despite the significantly reduced Mn/Fe ratio in the aboveground part under B treatment. The Mn/Fe ratio is the lowest under the optimal B concentration for plant growth [39,41,43].

Research has been conducted on the interaction between B and zinc (Zn) in plants. Overall, B accumulation has an antagonistic relationship to Zn accumulation [45]. B accumulation in plants will be increased under Zn deficiency [46,47], while it is decreased by the application of Zn fertilizers [26,48]. A declining Zn accumulation in *Z. mays* single-cross CS201 could be observed with an increase in B concentration [41]. Hence, the excess B toxicity to crops in cultivated land with high B concentration can be effectively relieved by the addition of Zn fertilizers [45]. Nonetheless, some research suggests that the application of B fertilizers to *N. tabacum* fails to reduce the accumulation of Zn [39]. Moreover, a synergistic interaction between Zn and B in *Brassica nigra* under extreme B deficiency and excess B toxicity has been reported, and the symptoms of Zn toxicity are aggravated by excess B [49]. In *Citrus sinensis*, B can alleviate the toxicity of excess copper (Cu) to plants by reducing the damage to roots and improving nutrient and water status [50].

B and Mo are two trace minerals that were recently identified to be essential for plants. Though research on them is still scarce, the interaction between B and Mo was uncovered in research on such crops as *B. napus*, *Bupleurum chinense*, and *Angelica dahurica*, and the combined application of B and Mo fertilizers can significantly improve the yield of crops compared with the application of B fertilizers alone [51,52,53]. This finding indicates the presence of an interaction between B and Mo.

### 2.6. B and Sodium (Na), Selenium (Se), and Silicon (Si)

Na, Se and Si are not considered “essential” elements for plants, but they may have benefits for growth or stress adaption of some plants at times. Salt stress caused by excess Na is now a major abiotic stress factor that threatens crop production. In soybean, exogenous Se, B and Se+B alleviated the salt-induced oxidative stress by enhancing the enzymatic activity of the antioxidant defense system and the glyoxalase systems [54]. A study on different Aegilops genotypes has shown that most of the genotypes revealed an increment in shoot Na uptake under B excess toxicity [55]. Considering the interaction of B toxicity and salinity stress, the term of “BorSal” was recommended for the combined B toxicity and high salinity state in the soil [56]. Numbers of genes were found to be expressed differently in between BorSal stress and individual boric acid or NaCl stress in several plants, such as *ZmPIP2* (*plasma membrane aquaporin 2*), *ZmPIP1*, *BOR* transporter genes in maize, *HvPIP1* and *HvPIP2* in barly, *NHX* transporter genes in *Suaeda glauca* [57,58,59,60].

B foliar spraying is an effective agronomic practice to cope with B deficiency in cotton. However, B has a narrow margin between deficiency and toxicity; thus, B spraying need to be well-controlled. De Souza Júnior et al. found that Si supplement reduced H_2_O_2_, primarily in B-deficient plants, and also increased proline and glycine-betaine concentration, mainly in plants under B toxicity. Therefore, adding Si to a B spraying solution for cotton plants can effectively mitigate the stress caused by either B deficiency or toxicity [61].

## 3. Interactions between B and Toxic Elements

Aluminium (Al) is the third most abundant element in the earth’s crust, but it is not essential for plant growth. Instead, it seriously threatens crop growth in acidic soils due to its higher solubility. The exogenous application of B fertilizers to a variety of plants can greatly alleviate the symptoms of Al toxicity, such as inhibited root growth, blocked photosynthesis, and impaired activity of relevant enzymes [62]. It is generally accepted that B can not only stabilize the cell wall structure, but also reduce the demethylation of pectin and lower the proportion of negatively charged carboxyl–carbonyl groups that can bind to Al by inhibiting the expression and activity of pectin methylesterase (PME) [63,64,65]. As a result, Al accumulation in the cell wall and Al absorption are reduced, and the toxicity of Al is alleviated. Different alleviation mechanisms of Al toxicity by B are observed in different plants. For instance, the application of B fertilizers to maize, *Cucumis sativus*, and *Poncirus trifoliate* can eliminate the oxidative stress induced by Al [66,67]. According to research on rice, B can regulate the expression of Al-related transporter genes, thus contributing to the compartmentalization and detoxification of Al into vacuoles [65]. The role of B in alleviating the Al toxicity to monocotyledonous plants was found to be limited in some research [65,68]. These findings cause no contradiction to the view that B can alleviate the Al toxicity to plants, since monocotyledonous plants have a lower demand for B [65].

The publication of such advancements as “cadmium (Cd) rice” has aroused the public concern about heavy metal pollution as an environmental issue, which poses serious threats to global food security and human health [69,70]. A series of recent studies have demonstrated that the application of B fertilizers can mitigate the Cd toxicity to crops and reduce the accumulation of Cd [71,72,73,74,75,76,77,78,79]. Compared with the case without B fertilizer application, Cd accumulation in the roots and aboveground part of rice is reduced by over 60% when 20–30 µmol·L^−1^ B fertilizers are applied. The Cd content is increased by 79% in the cell wall of leaf tissues but decreased by 64% in organelle components. Further research showed that B has an influence on components such as pectin in the cell wall, thus promoting the chelation of Cd by the cell wall, hindering the absorption of Cd, and improving the tolerance to Cd [76,78]. Moreover, the expression of Cd-related transporter genes in rice, such as *OsHMA2*, *OsHMA3*, and *OsNramp1*, is affected by the application of B fertilizers, which impede the transport of Cd to the aboveground part [71,78]. In addition, B can improve the tolerance of rice to oxidative stress resulting from Cd toxicity by promoting the activities of relevant antioxidant enzymes and reducing the concentrations of malondialdehyde, H_2_O_2_, and O^2−^ [71,76]. Similar results have been obtained from other crops, including *B. napus*, *Triticum aestivum*, and *Capsicum annuum* [72,73,74,77,79]. Hence, the negative correlation between B accumulation and Cd accumulation in crops provides a potential approach to the safe production of Cd-polluted crops.

## 4. Physiological Mechanism of the Interaction between B and Other Elements

### 4.1. Interaction with Other Elements via the Cell Wall

The cell wall serves as the first barrier for plant cells and a major site where plants store multiple elements, significantly influencing the absorption and transport of various minerals and toxic elements. As an important component of the cell wall in plants, B can influence the accumulation of and tolerance to other elements in plants by virtue of its effects on the structure and function of the cell wall, and this is considered to be an important way for B to interact with other elements (Table 1).

First, B can regulate the chelation of metal ions by the cell wall by influencing the charged functional groups associated with the cell wall. Divalent and trivalent metal cations can be bound to carboxyl groups (—COO^−^) and other negatively charged groups enriched in such components of the cell wall as pectin. Notably, their conjugation ability with metal ions is affected by the esterification degree of pectin [80,81]. Furthermore, other polysaccharide components, proteins, and phenolic substances also contribute to the chelation of metal ions by the cell wall [80]. The synthesis and esterification degree of pectin, as well as the contents of functional groups related to metal ion chelation, can be changed by B treatment [63,64,72,73,76,79]. Moreover, as observed in such crops as *B. napus* and *C. annuum*, B can increase the contents of cellulose, lignin, protein, and other components of the cell wall [73,77,78]. Under B treatment, the above physiological changes in the cell wall alter the chelation of ions including Cd^2+^, Ca^2+^, and Al^3+^ by the cell wall, ultimately affecting their absorption, accumulation and tolerance by plants.

There is an interaction between the B-mediated dimerization of RG-II in pectin in the cell wall and the accumulation of metal ions in plants. As found in research on a glycosyltransferase protein gene, *Cdi*, induced by the condition of Fe deficiency, the B-mediated A-chain galactosylation of RG-II plays a crucial role in the dimerization of pectin [44]. After Cdi mutation, the dimerization degree of pectin shows a sharp decline, and the chelation ability of cell wall to Fe is improved, thus hindering the reuse of apoplast Fe. The phenotype of hindered redistribution of apoplast Fe into the cells can be mitigated by the exogenous application of B fertilizers, but how B affects the chelation ability of the cell wall to Fe remains unclear [44]. Ca is another important player in the formation and stability of the dRG-II–B dimer. Ca^2+^ can form an “egg-box” structure by cross-linking with low-esterified homogalacturonic acid in pectin or a pectin network by combining with such complexes as dRG-II–B [82]. After the extraction of Ca from pectin, the dRG-II–B dimer is degraded and B is also released from the complexes [83], revealing the essential roles of both B and Ca in RG-II dimerization. Moreover, Ca^2+^ itself is involved in the formation of dRG-II–B complexes, and the role of exogenous application of Ca^2+^ in promoting the formation of dimers by B and RG-II has also been demonstrated in in vitro experiments [84,85]. Therefore, B may also interact with other elements like Fe and Ca by means of mediating the dimerization of RG-II to affect the cell wall structure.

### 4.2. Interaction with Other Elements through Competitive Inhibition

Many elements with similar chemical structures and properties, such as Cd and Fe, Mn, Ca, pentavalent arsenate and phosphate, share the same transport system in plants [86]. Therefore, competitive inhibition occurs during the absorption and transport of these elements and serves as a significant way for these elements to interact. Günes and Alpaslan found that P absorption by maize can be reduced by the exogenous application of B fertilizers and that B absorption by maize can be reduced by the application of P fertilizers, which indicates an antagonistic relationship between the absorption of B and P. They further proposed that in a high-pH environment, B is primarily absorbed by plants in the form of borate. Borate and phosphate have similar biochemical and physiological effects, possibly because of the closely associated absorption systems [26]. The antagonistic relationship between B and P has been observed in *S. lycopersicum* as well [25], but not in many other studies (as described above). It is worth pointing out that we have not identified any proteins involved in the transport of both borate and phosphate. Therefore, further research effort is still needed to explore whether the shared transport system is responsible for the competitive inhibition between B and P.

### 4.3. Interaction with Other Elements through Their Related Signal Transduction Pathways

Changes in the concentration of exogenous B can lead to variations in Ca^2+^ concentration and intracellular reactive oxygen species (ROS). Since Ca^2+^ and ROS are important signalling factors, the changes in B concentration may affect the expression of downstream genes through Ca2+ and ROS signal transduction pathways [38,87]. As observed in both Arabidopsis roots and N. tabacum BY-2 cells, B deficiency can induce changes in the gene expression related to Ca^2+^ signalling, thereby regulating the expression of downstream stress-responsive genes [87,88,89]. In research on *B. napus*, Zhou et al. proposed that B deficiency first induces the influx of Ca^2+^ and then triggers a ROS burst, and the Ca^2+^ signalling pathway functions in the upstream of ROS [87]. Although application of B has been observed to affect the expression of stress-responsive genes related to Al and Cd, further research is needed to find out whether B can regulate the expression of these genes by affecting Ca^2+^ and ROS signaling pathways.

## 5. Summary and Perspectives

B is an essential trace element for plants, and its reactions with other elements are widely recognized. Many of these interactions come from agricultural practice, namely, the combined application of B fertilizers and other fertilizers. In-depth research on the interactions between B and other elements will be important to guide the rational combination of B fertilizers and other mineral nutrients, thus avoiding the accumulation of harmful metal elements and improving the utilization efficiency of the nutrients. Current research on the interactions between B and other elements displays two notable characteristics. First, the influences of B on the accumulation and tolerance of deficiency/excess stress of different elements are not the same in most cases. Second, some inconsistent results have been obtained from different plants or different experiments, even under the influence of B on the same element. This might be related to the differences in experimental conditions and methods. In addition, extreme treatment conditions in experiments may lead to different results. For example, under extreme deficiency or excess B and other essential elements, the balance of elements in the plant is disrupted, thus potentially influencing all aspects of its normal life processes. In such cases, the observed interactions between B and other elements may not hold true. Moreover, these different observations also indicated that the interactions between B and other elements are complicated physiological processes.

The current understanding of B’s functions in plants mainly concerns its involvement in the structure and stability of plant cell wall. Thus, relatively more attention has been paid to the role of the cell wall in the interactions between B and other elements. Yet the physiological role and accumulation mechanism of B outside of the cell wall-related functions have been rarely explored, so it may be a direction worth exploring in future research on B nutrition. The method of forward genetics may be an effective way to find key genes in these unknown pathways. Recently, thanks to technological advancements and the integration of omics technologies, significant progress has been achieved in our understanding of the main quantitative trait loci related to B efficiency [90,91]. Since B is not directly involved in the cell wall composition of yeasts, we have obtained several plant genes mediating yeast tolerance to B stress by screening cDNA libraries of plants via yeast systems. The correlations of these genes with B stress have not been reported yet. These attempts will be of positive significance for expanding our knowledge of the new pathway of B metabolism in plants, and will lay a foundation for figuring out the molecular mechanisms of the interactions between B and other elements.

## Figures and Tables

**Table 1 genes-14-00130-t001:** Summary of interactions between B and other elements in plants.

Element		Interaction Pattern	Interaction Mechanism	References
Microelements	N	B promotes the absorption of nitrate by plants and its distribution in vivo. B promotes symbiotic N fixation in plants. Different forms of N sources affect the B absorption by plants	B can affect the expression of genes related to nitrate transport and metabolism. The ability of B to boost the symbiotic N fixation in plants may be related to its participation in maintaining the normal structure of the cell wall and cell membrane of root nodules, as well as in promoting root nodule infection and invasion. Different forms of N sources regulate B absorption by influencing the rhizosphere pH.	[14,15,16,17,18,19,20,21,22,23,24]
	P	In *S. lycopersicum* and *Z. mays*, there is an antagonistic relationship between the absorption of B and P. In *B. napus*, an appropriate amount of P can promote B absorption by plants.	Phosphate and borate may share the same absorption and transport system, presenting a competitive relationship. An appropriate amount of P may also promote the physiological processes of plants and improve the B availability in soil by affecting the biochemical characteristics of the plant rhizosphere.	[25,26,28]
	K	The accumulation of K and B in plants is mutually promoted.	Unknown	[32,39]
	Ca	The increase in Ca concentration aggravates the B deficiency symptoms of plants but alleviates the toxicity of high-concentration B stress. An appropriate Ca/B ratio is highly important for plant growth.	The interaction between B and Ca is related to the physiological processes of the cell wall, the Ca^2+^ signal transduction pathway, and the B availability in soil	[32,33,34,35,36,37,38]
	Mg	Mg accumulation in *S. lycopersicum* tissues is reduced with increasing B concentration, but changes in B concentration in *Z. mays* have no significant influence on Mg accumulation.	Unknown	[39,41]
	S	In *B. juncea* and *H. annuus*, the absorption of S and of B are mutually promoted.	Unknown	[42]
Trace Elements	Fe	B promotes the absorption and long-distance transport of Fe by plants.	B can affect the dimerization of pectin RG-II, thus regulating the chelation of Fe by the cell wall and influencing the reuse of Fe in root apoplast.	[43,44]
	Mn	B promotes Mn accumulation in plants.	Unknown	[39,41,43]
	Cu	In *C. sinensis*, B can reduce the absorption of Cu to alleviate the toxicity of excess Cu.	B can alleviate Cu damage to roots and improve nutrient and water status.	[50]
	Zn	There is an antagonistic relationship between the accumulation of B and of Zn in plants.	Unknown	[45,49]
	Mo	Combined application of B and Mo fertilizers can increase the yields of *B. napus*, *B. chinense*, *A. dahurica* and other crops	Unknown	[51,52,53]
Benificial Elements	Na and Se	B alleviates the salt-induced oxidative stress	B enhances the enzymatic activity of the antioxidant defense system and the glyoxalase systems.	[54,55,56,57,58,59,60]
	Si	Si mitigates the stress caused byboth B deficiency and toxicity	Si supplement reduced H_2_O_2_, primarily in B-deficient plants, and also increased proline and glycine-betaine concentration, mainly in plants under B toxicity.	[61]
Toxic Elements	Cd	B can reduce Cd accumulation in plants and alleviate the Cd toxicity to plants	B can promote the chelation of Cd by the cell wall, regulate the expression of Cd transport genes, and alleviate the oxidative stress caused by Cd.	[71,72,73,74,75,76,77,78,79]
	Al	B can reduce Al accumulation in plants and alleviate the Al toxicity to plants.	B can regulate the expression of Al transporter genes and pectin methylesterase genes and alleviate the oxidative stress caused by Al.	[62,63,64,65,66,67,68]

Note: Changes in the accumulation of other elements when plants grow abnormally due to an extreme deficiency of B or to the toxicity of excess B are not included.

## Data Availability

Not applicable.

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
