# Peer review of "Interaction between Boron and Other Elements in Plants"

_genes, 2023, doi:10.3390/genes14010130_

Round 1
Reviewer 1 Report
Boron is an essential element for plants and the interaction with other elements is a relevant and interesting topic.
In the introduction I have miss (1) information about Boron forms in the soil (boric acid, borate, etc) and the availability for plants in different types of soils; and (2) methodologies of how to quantify/follow the movement of B inside plant cells and tissues.
Each subsection of Section 2 should follow the same structure for the elements discussed: uptake (availability/forms in the soil/competition)/ transport (specific transporters/genes/signaling) / storage and function inside the cell, tissues and organs.
In 2.2. Develop more this statement: ‘phosphate and borate share the same absorption and transport system’ indicating the specific transporters/pumps.
In 2.3. ‘The Ca–B interaction has been verified by studies on numerous other plants’ needs a review citation.
In 2.5. ‘B can alleviate the toxicity of excess copper (Cu) to plants by downregulating the Cu absorption by plants’ explain how (the hypothesis of the authors).
Section 3 is well written and summarized recent research related to the topic.
Section 4:
I believe Na should be included, at least in the Summary Table 1. Have a look at for example: 10.3390/plants8100364
What about the paper of Boron in reproductive organs? Are there studies showing interaction of elements affecting developing tissues?
Minor comments:
Line 10: cadmium and aluminum
Line 17: change ‘was discussed’ by ‘has been reviewed’
Line 23: add verb: is needed to address the aforementioned
Line 31: remove ‘in plants’
Line 76: remove ‘First’
Ca2+ superscript
Line 159: N. tabacum italic
Line 210: remove “to”
Line 213: remove “of”
Reviewer 2 Report
Long and Peng have provided a review on the interaction of boron with other elements in plants. The topic is interesting, however, there are following crucial points that should be considered by authors to increase the value of the manuscript and may be readability.
- First of all, a lot of literature is available on boron and its interaction with other elements. Thus, it is necessary to discuss about the novelty of the manuscript in the abstract and introduction. The authors can do that by mentioning the deficiencies of the previous papers that this manuscript will fulfill.
- A most recent study on nutrient homeostasis discussed the interaction between boron and other elements, please read this manuscript (https://doi.org/10.3390/biology11081094). The results obtained in their study for various nutrients can be used to discuss the interaction between different elements present in your study.
- It will be good to add a figure showing the physiological mechanism of the interaction between B and other elements.
- As it is a review paper, it should cover the most important aspects of boron. I suggest adding a paragraph on both boron deficiency and boron toxicity to the article in the introduction. Although few lines are present on boron deficiency, boron toxicity is completely missing. Please add some information on toxicity. Authors can add the information on boron toxicity from these latest papers and cite them (https://doi.org/10.3389/fpls.2021.736614; https://doi.org/10.3390/agronomy12102421).
- The manuscript is submitted in the journal genes. I suggest authors to add some information on the genes related to interaction between boron and other elements in a separate paragraph.
- Similarly, interaction of boron and sodium is very well known for reducing the effects boron stress. Please add a paragraph on their interaction in section 2. The authors can take the information from the following manuscript and cite it (https://doi.org/10.3390/plants8100364).
I do believe that the manuscript can be better once the authors address the mentioned points and enrich the manuscript with crucial information.
Reviewer 3 Report
The manuscript presents a review on the interaction between boron and other elements in plants. The manuscript is worth reading but needs major revisions. As this scientific topic needs so much research and experiments, it will be potentially interesting to arrange a review on this. However, it also includes the responsibility to cover all the important aspects of the topic, and, compile and discuss the most recent articles on the topic. The authors can cite these most recent articles and add their information to the paper, it will increase the value of the manuscript (https://doi.org/10.3390/plants10102224; https://doi.org/10.3390/plants11212937; https://doi.org/10.3390/biology11081094; https://doi.org/10.1186/s12870-022-03721-7). Moreover, though the authors have tried to cover most of the element's interaction, sodium, silicon, and selenium have not been included. A figure can be added to the manuscript.
Round 2
Reviewer 1 Report
The authors have properly addressed and replied the main concerns exposed by the reviewers.
Reviewer 2 Report
-
Reviewer 3 Report
The authors have tried to incorporate the suggestions and the manuscript can be accepted in its present form.